# Continuous-variable tomography of solitary electrons

J.D. Fletcher [1], N. Johnson [1,2,6], E. Locane[3], P. See[1], J.P. Griffiths[4], I. Farrer [4,7], D.A. Ritchie [4], P.W. Brouwer[3], V. Kashcheyevs [5] & M. Kataoka [1*]

A method for characterising the wave-function of freely-propagating particles would provide a useful tool for developing quantum-information technologies with single electronic excitations. Previous continuous-variable quantum tomography techniques developed to analyse electronic excitations in the energy-time domain have been limited to energies close to the Fermi level. We show that a wide-band tomography of single-particle distributions is possible using energy-time filtering and that the Wigner representation of the mixed-state density matrix can be reconstructed for solitary electrons emitted by an on-demand single-electron source. These are highly localised distributions, isolated from the Fermi sea. While we cannot resolve the pure state Wigner function of our excitations due to classical fluctuations, we can partially resolve the chirp and squeezing of the Wigner function imposed by emission conditions and quantify the quantumness of the source. This tomography scheme, when implemented with sufficient experimental resolution, will enable quantum-limited measurements, providing information on electron coherence and entanglement at the individual particle level.

[1] National Physical Laboratory, Hampton Road, Teddington, Middlesex TW11 0LW, UK. [2] London Centre for Nanotechnology and Department of Electronic and Electrical Engineering, University College London, Torrington Place, London WC1E 7JE, UK. [3] Dahlem Center for Complex Quantum Systems and Institut für Theoretische Physik, Freie Universität Berlin, Arnimallee 14, 14195 Berlin, Germany. [4] Cavendish Laboratory, University of Cambridge, J. J. Thomson Avenue, Cambridge CB3 0HE, UK. [5] Department of Physics, University of Latvia, Jelgavas street 3, Riga LV 1004, Latvia. [6] Present address: NTT Basic Research Laboratories, NTT Corporation, Atsugi, Japan. [7] Present address: Department of Electronic & Electrical Engineering, The University of Sheffield, Mappin Street, Sheffield S1 3JD, UK. *email: masaya.kataoka@npl.co.uk

nitializing and measuring the wave-function of single freely-propagating particles are challenging but fundamental tasks for applications in quantum information processing and enhanced sensing[1–10]. The recent development of semiconductor-based single-electron sources[11–13] and ways of controlling electron propagation[14–16] have created a new platform harnessing on-demand electronic excitations in this way. These schemes for single electron quantum optics require techniques to both control and probe the single excitations. Specific properties of the sources and the transmission channels into which excitations are launched give rise to different characteristic excitation energy, ejection dynamics, propagation velocity and interactions. As a result, new methods are demanded for reconstruction of the quantum state in different systems[17,18].

The key properties of the emitted electron stream are manifest in the first-order coherence, captured by Wigner quasi-probability function $W(E, t)$. The Wigner function $W(E, t)$ is not directly measurable, but projections along specific trajectories in the phase space of non-commuting variables (position–momentum, energy–time) can be accessed, enabling a tomographic reconstruction[19] somewhat like X-ray tomography. Such measurements require a scheme to create and readout projections at different trajectories or mixing angles, for instance via free space evolution of the transverse wavefunction of atomic beams[20,21] or by mixing of photons with a local optical field[4,22]. In this way continuous-variable quantum tomography techniques, developed for atomic beams[20] and photonic modes[23], have been successfully adapted for on-demand electronic excitations[17,24,25].

In the case of a chiral one-dimensional electronic excitations the Wigner function can be written as

$$W(E, t) = \frac{1}{h} \int e^{it\epsilon/\hbar} \langle E - \epsilon/2 | \hat{\rho} | E + \epsilon/2 \rangle d\epsilon \qquad (1)$$

where $\hat{\rho}$ is the density matrix of the emitted electrons and the energy eigenstates $|E\rangle$ form a complete basis for the propagating mode. The Wigner function of low energy excitations can be extracted by using two-particle interference of the electron beam with a modulated Fermi sea as a local oscillator[17,18]. This is only possible in a restricted phase-space volume close to the Fermi energy ($E - E_F \ll 1$ meV) and is not viable over a wider range of parameters, such as excitations in a higher energy range[15]. It is also not possible where there is no Fermi sea, as in the case of isolated electrons travelling in a depleted lattice space without conduction-band electrons nearby[15,26]. However, in these cases it is possible to interrogate the beam with a barrier in the beam path[13,15,27], an approach which can enable a different method of tomographic reconstruction of the Wigner function[28].

Here, we explore a tomographic technique to image the distribution of Wigner quasiprobability for electrons in phase space reconstructed from a set of projections acquired by energy and time selective transmission. The selective control of transmission is achieved by a dynamic barrier in the beam path synchronised with electron emission. This is a technique that is applicable over a wide range of energy and time scales. We use this approach to perform tomography of electrons emitted by an on-demand single-electron source, enabling us to directly characterise the energy-time distribution of excitations at a particular point in a beam path. Using the phase space density we quantify the quantum mechanical purity of the states. We also demonstrate readout and control of an important signature of ejection dynamics, a chirp due to correlation between arrival energy and time, which illustrates the power of the technique in controlling single electronic excitations.

## Results

**Electron tomography using time-dependent barriers.** Marginal distributions at different projection angles in energy–time space can be measured using interaction with a time-dependent barrier in the beam path[28]. We measure the transmission probability $P_T$ for electrons filtered by a high-pass energy barrier with a linearly driven time-varying transmission threshold $E_T(t) = E_{T0} + \beta_E t$ as in Fig. 1a. The connection to the Wigner function $W(E, t)$ is established via

$$P_T = \iint W(E, t) \, T[E - E_T(t)] \, dE \, dt \,, \qquad (2)$$

where $T[E - E_T(t)]$ can be interpreted as the time- and energy-dependent transmission quasiprobability of the barrier, which masks part of the Wigner distribution as in Fig. 1a.

For an intuitive understanding of our tomography protocol it is convenient to use polar coordinates $\theta, S$ as shown in Fig. 1a and methods. The sweep rate $\beta_E$ sets a projection angle $\theta$ in the energy time plane via $\tan\theta = \beta_E/\beta_0$ ($\beta_0$ sets the energy/time aspect ratio). For a sharp threshold barrier with $T(E) = 1$ or 0 for $E > 0$ or $E < 0$, respectively, the derivative $dP_T/dS$ is proportional to the integral of $W(E, t)$ along the line $E_T(t)$. The line $S$ (indicated in Fig. 1a) is perpendicular to the $E_T(t)$ line and is akin to the detector coordinate in an X-ray tomography scheme. By measuring $dP_T/dS$ systematically for various values of $\beta_E$ and $E_{T0}$ (to select the position along $S$) we obtain the Radon transform (sinogram) of $W(E, t)$. Its reconstruction is then possible using filtered back-projection to implement the inverse Radon transform (see methods).

**Experimental scheme.** Our device components, electron source and energy barrier, are defined by gates on a GaAs-based heterostructure[26], as shown in Fig. 1b. A tunable-barrier electron pump[29,30] (left hand side) is operated using a periodic voltage $V_{G1}(t)$ (Fig. 1c, left hand side) applied to the left-most barrier, pumping one electron per cycle through the device at a repetition rate $f$ giving a quantised pump current $I_P = ef$[29,31–34]. The right hand barrier, controlled by $V_{G2}$ determines the number of electrons pumped, and linearly controls the ejection energy[15] (see Supplementary Fig. 1). After ejection, each electron follows a trajectory along the mesa edge governed by the side-wall potential and Lorentz force due to an externally applied perpendicular magnetic field $B$ until it reaches the potential barrier controlled by voltage $V_{G3}(t)$ (Fig. 1c, right hand side)[26]. The barrier control voltage $V_{G3}(t)$ is synchronised to $V_{G1}(t)$ with adjustable delay $t_d$[27].

The edge gate (which depletes the region of carriers for negative gate voltages $V_{G4}$[26,35]), the injection energy (far above the Fermi energy—see Supplementary Note 1), and the travelling time and operation frequency (transit time much shorter than pumping repeat time) lead to one isolated electron being present in the edge channel at a time[26]. In our implementation of the tomography scheme, $V_{G3}(t)$ controls the energy threshold $E_T(t)$ and therefore determines what proportion of the pumped current is transmitted $P_T = I_T/I_P$ (see methods). We use an arbitrary waveform generator to control the threshold barrier time dependence, which we set to have an adjustable linear ramp rate $\beta_E = -\alpha_h dV_{G3}(t)/dt$ near the moment of electron arrival, where $\alpha_h = (0.61 \pm 0.02)$ meV/mV (see Supplementary Note 1 and Supplementary Fig. 1) and $\beta_0 = (0.12 \pm 0.01)$ meV/ps (see Supplementary Note 3 and Supplementary Figs. 3, 4). We then shift the transmission mask in increments $\Delta S$ along the $S$−axis using a combination of time delay $t_d$ and DC voltage shift $V_{G3}^{DC}$ (which controls $E_{T0}$) for each angle $\theta$ and measure the

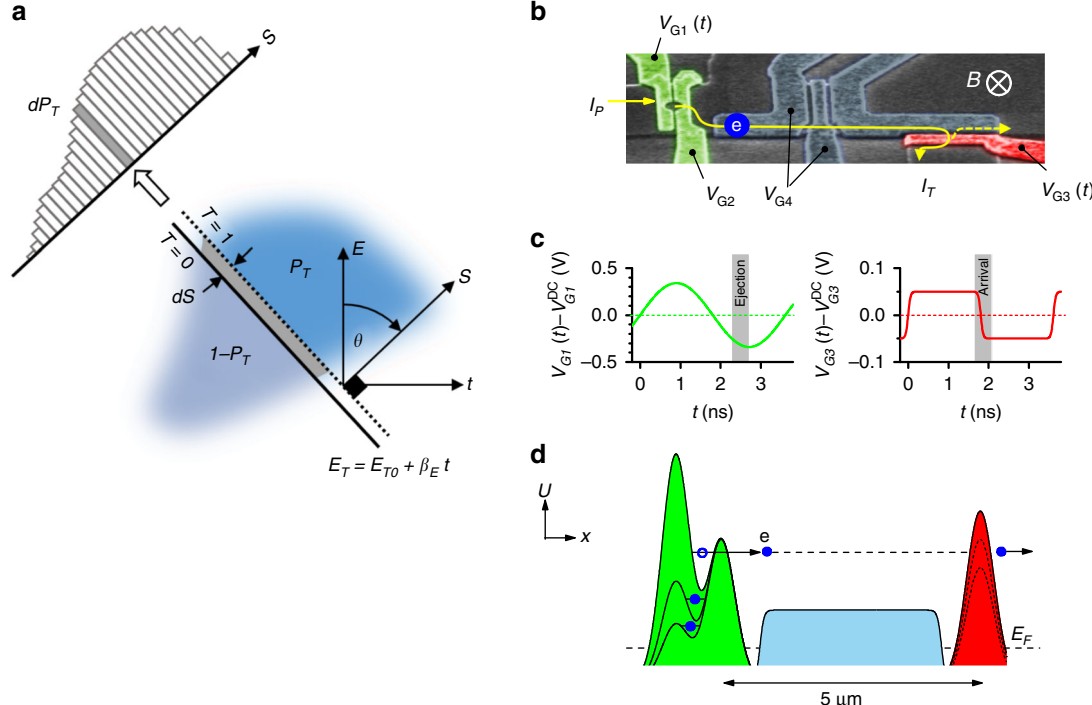

**Fig. 1** Electron tomography scheme using a modulated barrier. **a** An unknown Wigner distribution $W(E, t)$ of a periodic electron source electron can be filtered using a linear-in-time threshold energy barrier set at height $E_T$. The transmitted and reflected part, labelled $P_T$ and $1 - P_T$ result in a proportionate transmitted and reflected currents. A marginal projection of this distribution in the energy, time plane can be measured by fixing the ramp rate of the barrier $\beta_E$, which sets $E_T$, then moving the threshold boundary along the axis $S$ in increments $dS$, while measuring the resulting changes in transmitted current. Repeating the experiment at different ramp rates (which sets the angle $\theta$) gives enough information for a numerical reconstruction of the distribution. **b** False-colour scanning electron micrograph of device identical to that measured (see methods for details). The electron pump (left, highlighted green) injects pump current $I_p$. The barrier (right, highlighted red) selectively blocks electrons giving transmitted current $I_T \leq I_P$. The path between these is indicated with a line. The gates along the path (controlled by $V_{G4}$) depletes the underlying electron gas but do not block the high energy electrons. **c** Typical time-dependent control voltages for pump $V_{G1}$ and probe barrier $V_{G3}$ (each has a DC offset—see methods). **d** Electron potential $U(x)$ along the electron path between source and probe barrier at three representative stages for pumping (left) and blocking (right).

transmission probability changes from the change in transmitted current $\Delta P_T = \Delta I_T / I_P$.

**Sinogram and tomographic reconstruction.** The numerical derivative $\Delta P_T / \Delta S$ collected at different angles $\theta$ (a sinogram) is shown in Fig. 2a. Each cut represents a projection of the distribution at a different angle, as indicated in Fig. 2b–d. At $\theta = 0°$ this shows the distribution along the energy axis (as in Fig. 2c) having a width of order ~2 meV[15]. At the highest angles $|\theta| \rightarrow 90°$ (notwithstanding some minor bandwidth limitations, see Supplementary Note 3) this maps the time of arrival distribution (Fig. 2b) with a width of order $~10^{-11}$ s[36]. At intermediate angles, for instance at $\theta = 60°$ in Fig. 2d the cut contains a mixture of energy and time information.

We use filtered back-projection[23] to compute the electronic distribution from this sinogram, as shown in Fig. 2e. This method enables reconstruction of the mixed state Wigner distribution of electrons, a combined map of electron energy and arrival time measured relative to the centre coordinates used for the collection of the sinogram data (here aligned to the mean electron energy and arrival time $E_0$ and $t_0$). As some classical fluctuations are present in our experimental implementation we interpret our results as an effective mixed state Wigner function[28]. Our results are therefore an ensemble measurement of many pure states, with a lower phase-space density than a pure state (discussed below). We can, however, resolve a feature likely to be common to each pure state, a feature of the distribution that we show can be controlled by electron ejection conditions.

**Effect of electron ejection dynamics.** We observe that measured phase space distribution Fig. 2e is stretched at a certain angle in the energy–time plane, a feature derived from the sharpening of the projection at the corresponding angle in Fig. 2a. This chirp (i.e. time-varying frequency/energy) of the arriving electron energy distribution is an expected feature of electron ejection from a quantum dot under non-stationary conditions; in this kind of pump the driving barrier forces ejection by raising the dot energy with respect to the exit barrier[29,37]. Previous experiments showed hints of energy–time correlation[36] but this is now directly visible in the measured distribution. Indeed, we can show that the chirp can be controlled by changing the conditions under which the electron is ejected.

The entrance barrier DC control voltage $V_{G1}^{DC}$ influences when electrons are ejected within the pump waveform[27]. For different values of $V_{G1}^{DC}$ within the $I_p = ef$ plateau (see operating points in Fig. 3a) the arrival time $t_0$ can be adjusted over a range of more than 400 ps (~11% of the total pump cycle time), as shown in Fig. 3b (circles). For our sinusoidal pump waveform, the sweep rate near the point of ejection is also tuned over a wide range (Fig. 3b squares). This can be set from a maximum rate of $|dV_{G1}/dt| \simeq 0.5$ mV/ps to ejection under almost static conditions $dV_{G1}/dt \rightarrow 0$. Reconstructions of the Wigner function at these operating points are shown in Fig. 3c. These show that the energy–time correlation is controlled by ejection speed. From fits to the data (see Supplementary Note 5) we find that energy–time trajectory $d\langle E \rangle / dt$ (see Fig. 3d) tracks the sweep rate of the pump drive barrier, with an estimated strength of the coupling between

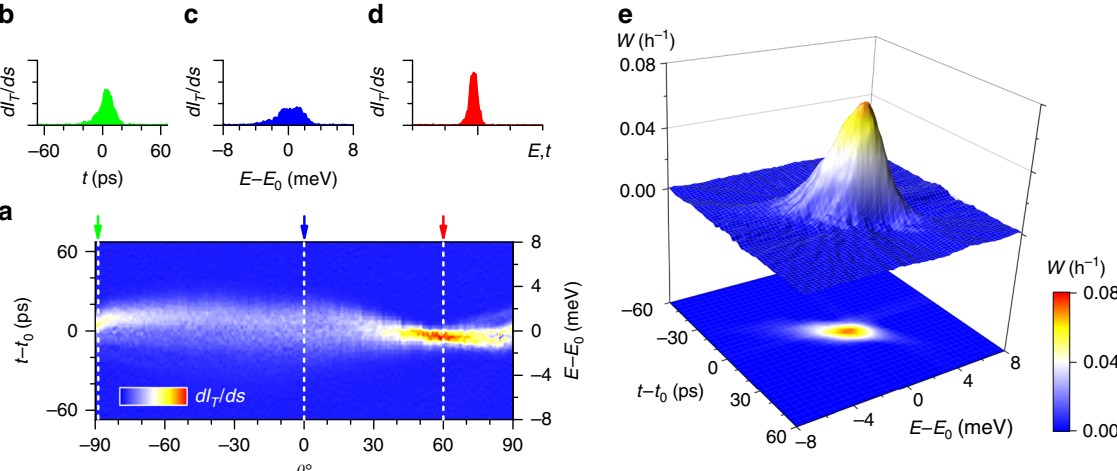

**Fig. 2** Angle dependent projection of single electron density. **a** Colour plot: projected electronic density (sinogram) at various angles $\theta$ in the energy–time plane. Colour scale corresponds to $dI_T/dS$, where $dS$ is an incremental step in the (normalised) energy–time plane. Depending on $\theta$, the projection axis $S$ corresponds to an energy projection, time projection or a mixture. The left axis is appropriate for $\theta = \pm 90°$ and the right hand for $\theta = 0°$. Selected projections are shown at angles, where $S$ corresponds to **b** time projection, **c** energy projection and a mixed projection **d**. **e** Inverse Radon transform of the data in **a** giving the Wigner phase-space density in units of $h^{-1}$.

the pump drive barrier and the emission energy of $d\langle E\rangle/dV_{G1} \simeq 0.41$ meV/mV.

**Resolution and the quantum limit**. Under real experimental conditions, the measured distributions can deviate from that expected of a pure quantum state. Additional broadening is expected if the emitted state is mixed, something that can be quantified using the measured phase space maps. A conservative measure of quantum indistinguishability of our wave-packets, relevant for two particle interference experiments, is the maximal statistical weight of a pure state in the mixture[18], $P_1 = \max_\psi \langle \psi|\hat{\rho}|\psi\rangle$. Values of $P_1$ can be obtained by numerical diagonalization of $\hat{\rho}$ or, for the range of values found here $P_1 \ll 1$, $P_1$ has a simple relationship to peak value of the measured phase-space density $P_1 \approx hW_{max}$ (see Supplementary Note 6 and Supplementary Table 1 for a comparison). We find $P_1 = 0.02$–$0.07$ as plotted in Fig. 3e (solid symbols). Similarly, the purity of the effective mixed state, $\gamma = \mathrm{Tr}(\hat{\rho}^2)$, is related to the average phase space density of the Wigner representation, $\gamma = h\langle W(E,t)\rangle = h\iint W^2(E,t)dE\,dt$. From the data in Fig. 3c we find $\gamma = 0.01$–$0.04$.

Our Wigner function reconstructions may also be influenced by certain experimental limitations, for instance the energy broadening of the barrier transmission. For a monotonic detector barrier transmission function $T(E)$ going from 0 to 1 over a finite energy scale $\Delta E$ is equivalent to smoothing of the underlying $W(E,t)$ by a convolution with $dT/dE$ along the energy axis, resulting in a smeared density distribution and reduced values of $P_1$ and $\gamma$[28]. A semiclassical model calculation[38] using $\Delta E = 0.8$ meV, the narrowest energy feature seen experimentally (see Supplementary Note 4), suggests a maximum measurable phase space density $hW_{max} \lesssim 0.16$ (Fig. 3e open symbols). Our ability to rotate the source electronic distribution as in Fig. 3c enables us to further probe temporal and energy resolution limits and combine these in an estimate of our experimental resolution. A conservative estimate of the areal resolution is the product of the minimum projected energy width $\sigma_{E,min} \simeq 0.8$ meV (under slow ejection conditions) and the minimum projected temporal width (under fast ejection conditions) $\sigma_{t,min} \simeq 5$ ps, giving $\sigma_{E,min}\sigma_{t,min} \simeq 6.1\hbar$. While this is larger than the absolute minimal level of quantum uncertainty $\hbar/2$, this is an upper limit and is also clearly sufficient to resolve non-trivial properties of the excitations studied here. An estimate of the

temporal resolution limit from the maximal barrier sweep rate $\sigma_t' = \sigma_{E,min}/(\alpha_h dV_{G3}^{AC}/dt) \simeq 0.3$ ps gives $\sigma_{E,min}\sigma_t' \simeq 0.36\,\hbar$, suggesting that observation of higher purity states than that seen in this source may be possible in our scheme. How details of exact barrier geometry control this resolution limit, and the correspondence of this to the one-dimensional scattering problem[28] are open to further detailed study.

## Discussion
The ability to tune and readout the properties of electron sources is a potentially useful tool. For instance, periodic electron sources can act as a sensitive probe of on-chip signals[39], with an energy–time resolution set by the electronic phase space distribution. Similar to squeezed states in photonics[4], it should now be possible to enhance the resolution of measurements along certain phase-space trajectories. In situ Wigner function read-out will also aid the development of electron quantum optics devices where precise control of the Wigner function is required[11]. It should also be possible to use this scheme to detect coherences via negative fringes in the Wigner function arising from interference effects[28,38]; the characteristic oscillation period estimated from the kinetic energy of drift motion[26] is $\simeq 2$ ps, close to our accessible bandwidth.

In summary, we have shown a technique of generalised electron quantum tomography using numerical back-projection. Our method can reveal non-trivial emission distributions arising from internal dynamics of the quantum dot. The average and the maximal phase space density are 4 and 7% of the quantum limit, which is partly explained by finite resolution effects, but observation of a quantum-limited Wigner function should be possible.

## Methods
**Quantum tomography scheme**. Our experimental implementation maps closely onto a model of scattering between 1-dimensional chiral edge channels under a dynamic barrier which gives Eq.(2) as a result[28]. While a similar expression has been derived in the classical limit[36], this differs in microscopic approach and in the physical meaning attributed to its components (e.g. a classical joint probability distributions versus the Wigner quasiprobability). The model of ref.[28] includes physical effects that we expect experimentally; modification of electron energy by the barrier itself is explicitly included, and the non-trivial geometry of the barrier edge is considered (experimentally this is not infinitely sharp). The derivation also carries through with no correction in a fully quantum mechanical treatment as we outline here.

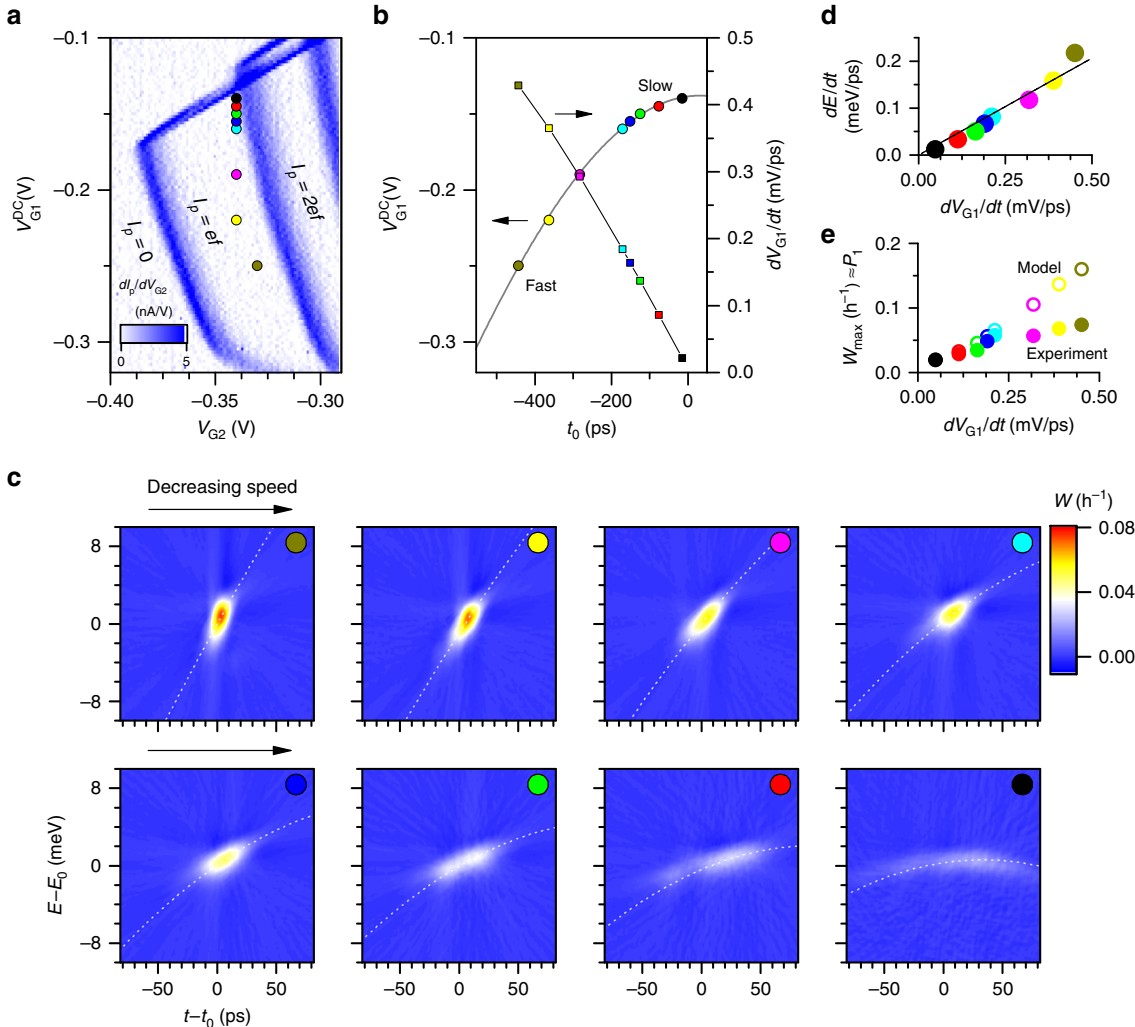

**Fig. 3** Tomography of excitations produced under different ejection conditions. **a** Coloured points show different operating points within the one electron/cycle pump current plateau whose boundaries are visible in this plot of $dI_p/dV_{G2}$. **b** Circular symbols indicate measured time of arrival, $t_0$ at each of the $V_{G1}$ operating points in **a**. Square symbols show the estimated barrier sweep rate at these times. **c** Single electron tomography for each pump operating points in **a**, as indicated by the coloured dots (top left is the fastest sweep rate, bottom right is the slowest). Dashed lines are the ejection trajectories calculated with a semiclassical model (See Supplementary Note 4 and Supplementary Figs. 5–7). **d** single electron chirp rate $dE/dt$ as measured from fits to the backprojection at each sweep rate on the pumping gate (the solid line is a linear fit). **e** Solid symbols are the measured peak phase space density $\approx P_1$. Open symbols are from a model that accounts for energy broadening.

Central to the approach is the observation that the transmission probability in the presence of a purely linear-in-time modulated voltage is equivalent (by gauge invariance) to transmission through a static barrier of a wave-packet with an additional quadratic phase factor[28]. One can choose a gauge, in which the electron energy is measured with respect to the transmission threshold energy, set by a barrier height. The presence of the barrier shifts the incoming energy–time distribution along the energy axis as the electrons lose momentum upon entering the gate-affected region. However, if the gate voltage (and hence the decelerating force) depends on time, then the incurred energy shift will depend on the arrival time too, thus deforming the energy–time distribution as it enters the barrier region. For the special case of linear-in-time modulation of the gate, this has a simple shift-and-skew effect on the distribution, which is then filtered at a constant threshold. This maps exactly to the selective transmission effect of Fig. 1a. It also turns out to be independent of the exact spatial profile of the gate edge potential leading into the scattering region, down to some small constant energy and time offsets that reflect the effective position of the barrier edge[28].

More specifically, for a static barrier, the probability of transmission is expressed quantum mechanically as

$$P_T = \int |\psi_{\text{out}}(t)|^2 dt = \iint T(E) W(E,t) dE\, dt , \qquad (3)$$

where $T(E) = |\tau(E)|^2$ is the square of a complex scattering amplitude $\tau(E)$ that connects the incoming and the outgoing probability amplitudes, $\psi_{\text{out}}(E) = \tau(E)\, \psi_{\text{in}}(E)$, and $W(E,t) = h^{-1} \int \psi_{\text{in}}^*(E+\epsilon/2)\psi_{\text{in}}(E-\epsilon/2) e^{i\epsilon t/\hbar}\, d\epsilon$

is the Wigner function of the incoming (pure) state. A uniform energy modulation of the whole scattering region, as in the case of the time dependent barrier height, can be expressed as a global energy shift $E \rightarrow E + E_{T0} + \beta_E t$, where $E_{T0}$ is an adjustable offset and $\beta_E$ controls the ramp speed. This is equivalent to a gauge transformation $\psi(t) \rightarrow \psi(t) e^{i(E_{T0}t + \beta_E t^2/2)/\hbar}$ where $\psi(t) = h^{-1/2} \int \psi(E)\, e^{-iEt/\hbar} dE$, which in turn leads to $W(E,t) \rightarrow W(E + E_{T0} + \beta_E t, t)$ in Eq. 3, and hence to Eq. 2.

Linear-in-time barrier sweeps (as used experimentally) should ensure that the electronic distribution after entering the time-dependent barrier region remains undistorted (down to the energy shifts described above) regardless of the spatial barrier shape (e.g. onset sharpness, overall size) $V(x,t)$. See ref. 28 for more detailed discussion of model approximations and practical constraints in the quantum limit. For discussion of the range of experimental applicability of this technique (e.g. energy range, effects of available experimental bandwidth) see Supplementary Note 7.

**Device design and operation.** Our device is defined by surface gates on a GaAs-based two dimensional electron gas heterostructure 90 nm below the surface[26]. Distance between the electron pump and energy-selective barrier is ~5 μm, as estimated from lithographic dimensions. The device is operated in a dilution refrigerator with base temperature ~100 mK (with RF drive signals turned on) in a perpendicular magnetic field $B = 12$ T. The wafer carrier density is ~$1.7 \times 10^{15}$ m$^{-2}$ with mobility 170 m$^2$ V$^{-1}$ s$^{-1}$. This carrier density and field places the bulk filling factor $\nu < 1$, but this is of secondary importance here because of the large energy and spatial separation between our excitations and the Fermi sea, due to the high electron

energy and the depletion gate $V_{G4}$ (see below for details). This is the same kind of device as used to measure electron velocity[26] and phonon emission[35].

**DC current measurements**. DC current readings are taken with commercial transimpedance amplifiers at $10^{10}$ V/A gain. Amplifiers are connected on the pump ($I_P$) and on the far side of the energy-selective barrier ($I_T$) as shown in Fig. 1 b. Although not used in our analysis, we also measure $I_R$, the reflected current with a third amplifier to confirm that the pumped current is divided between the two output terminals i.e. $I_P = I_T + I_R$ (for simplicity we consider electron current rather than conventional current). During tomography measurements, each current measurement (lasting 200 ms) corresponds to $\sim 5.5 \times 10^7$ pump cycles, while every back-projection map samples a total of $1.5 \times 10^{12}$ pump cycles.

**RF connections**. The waveforms are synthesized using two Tektronix 70001A Arbitrary Waveform Generators (AWG) connected via a synchronisation unit to effectively give two outputs, one for the source and the other for the energy selective barrier. Both RF signal paths use low-loss cryogenic coaxial cable (beryllium copper, superconducting) inside the cryostat. Broadband (18 GHz) 3 and 1 dB attenuators are present on both lines inside the dilution refrigerator for thermalisation purposes, in addition to 3 dB at room temperature. Due to the larger amplitude requirements for the pump, this line includes a 15 dB linear amplifier (15 GHz bandwidth). Broadband bias tees (18 GHz bandwidth) are used to add DC voltages at cryogenic temperatures near the sample. A 6 GHz low-pass filter was used on the pump drive signal to prevent weak oscillations creating ejection from a non-monotonic drive signal[27].

**Pumping**. A periodic voltage $V_{G1}(t) = V_{G1}^{AC}(t) + V_{G1}^{DC}$ [Fig. 1b, left] is controlled by one AWG channel and a DC voltage source. $V_{G1}^{AC}(t)$ and $V_{G1}^{DC}$ are the ac and dc components. The ac component modulates the G1 barrier, pumping $n$ electrons per cycle through the device at a repetition rate $f = 277$ MHz, giving $I_P = 44.4$ pA for $n = 1$. The tunnelling processes which select the number of loaded electrons have been discussed extensively in the context of accurate current standards for metrology[30–32]. Note that in the last panel in Fig. 3(c) the escape rate is reduced such that the electron cannot fully escape within the time permitted by the pump waveform, reducing the pump current by $\sim 8\%$.

**Waveform synthesis and delay control**. Both $V_{G3}(t)$ and $V_{G1}(t)$ waveforms are $N = 180$ points long, 10-bit vertical resolution and played cyclically at a frequency of $f_0 = 277$ MHz. The phase $t_d$ of the two sources is controlled by phase-shifting their synchronisation clock. The pump drive is a sine wave, while $V_{G3}(t)$ is of the form $v_k = \tanh[A_0 \tan(\theta) \sin(2\pi f_0 t_k)]$ for each time $t_k$. $\theta$ is the required projection angle and $A_0$ linearly scales the slope. This gives a linear voltage ramp near the zero crossings, while smoothly limiting the signal away from this point (see Supplementary Fig. 3). The actual sweep rate was measured in situ[39] (see Supplementary Fig. 4) for different values of $\theta$ and $\beta_0 = 0.12$ meV/ps was empirically found using $\tan \theta = \beta_E / \beta_0$ (i.e. $\beta_0$ is the sweep rate at $\theta = 45°$). The zero crossing of $V_{G3}(t)$ are matched to the electron arrival energy and time using the DC offset $V_{G3}^{DC}$ and the time delay $t_d$. Precise alignment is possible because the waveform, including the linear ramp region, is apparent in a map of transmitted current as described previously[15,39]. This accounts for RF cable length, the position of the ejection point in the pump waveform and the time for the electrons to traverse the device. The electron velocity measured in similar devices[26] is $\sim 0.5$–$1.5 \times 10^5$ ms$^{-1}$ giving an expected transit time $\sim 30$–$100$ ps[13,15,26,27].

**Backprojection**. Ideally, data would be collected by controlling the transmission mask via combined shifts in $\Delta t_d$ and $V_{G3}^{DC}$ (along the axis $S$) while using the angular control of the voltage sweep-rate to define the projection angle $\theta$. In practice it is difficult to collect data along arbitrary axis $S$ (because of finite resolution in voltage and time delay controls) so we measure along a convenient $S'$ and project this onto the $S$ axis (see Supplementary Note 2 and Supplementary Fig. 4). We use a standard procedure for the inverse Radon transform[23] with a ramp (high-pass) filter before numerical back projection (we also include a Hann low-pass filter for some noise rejection) (see Supplementary Note 2 and Supplementary Fig. 2).

## Data availability

The experimental data that support the findings of this study are available in the SEQUOIA community repository at https://zenodo.org/communities/sequoia/ at https://doi.org/10.5281/zenodo.3533120.

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

## Acknowledgements

We acknowledge valuable discussions with Heung-Sun Sim. We acknowledge use of software developed by Franz Ahlers. This work was supported by the UK government's Department for Business, Energy and Industrial Strategy and partly from the Joint Research Projects 15SIB08 e-SI-Amp and 17FUN04 SEQUOIA from the European Metrology Programme for Innovation and Research (EMPIR) cofinanced by the Participating States and from the European Unions Horizon 2020 research and innovation programme. We acknowledge support by the SFB 658 of the DFG and University of Latvia grant no. AAP2016/B031.

## Author contributions

J.D.F. performed the measurements and data analysis. N.J. assisted J.D.F. with the measurements. E.L., P.W.B., V.K. formulated analytical model for the transmitted current through a time-dependent barrier. P.S. fabricated the sample. J.P.G. performed e-beam patterning. I.F. and D.A.R. provided the wafer. M.K. developed the idea for a classical tomography (expanded by V.K., E.L., P.W.B. to a quantum tomography), designed the sample and led the project. All authors contributed to writing the paper.

## Competing interests

The authors declare no competing interests.
