## [Peer Review File · Nature Communications]

Reviewers' comments:

Reviewer #1 (Remarks to the Author):

This article reports on a new tomography techniques applied to a single electron source in the quantum Hall regime.

This experiment is quite different from previous tomography experiments (e.g. the one reported in D.C. Glattli group) that discuss small energy excitations. Here an electron at high energy $E_0 \sim 100\text{meV}$ is sent into the system.

I think that the experiments as such are sound and of high quality. The text is also clear. However, I am exceedingly puzzled by the presentation that is, if I understood correctly, very misleading.

The name "Quantum Tomography" implies getting access to some quantum aspect of the electron. Quantumness is also implied by sentences such as "Our method can be used to test the degree of 'quantumness' of an electron source from measurements of the phase- space density" or "the resolution should be sufficient to enable quantum-limited measurements" although in the last sentence, the usage of the word "should" seems to imply that this goal has not been reached yet.

On the other hand, unless I am mistaken - I would be happy to be contradicted - I feel that the observed data can be entirely understood at the classical level (e.g. the classical model of Fig. S4b looks quite good when compared to Fig.3) which is consistent with my understanding of what is going on. Indeed, measuring "quantumness" would require accessing the very fast oscillations of the wavefunction which occur at very fast time scale (10^{-14} seconds approximatively for 100meV) which sounds very difficult with a setup that measures at frequencies smaller than 100 GHz. Unless the authors can show that coherent off-diagonal elements of the density matrix are present (negative values of the Wigner function), there is no convincing evidence for "quantumness".

Hence, at minima, I recommend that the manuscript be striped from all the misleading quantum aspects (or that the authors provide convincing proof that a quantum treatment is needed to explain the data). It does not follow that the results are uninteresting, but a proper discussion of the new information conveyed on the classical probability distribution would be needed to assess this aspect. I do not recommend publication of this manuscript in its present state.

Reviewer #2 (Remarks to the Author):

The article "Quantum tomography of Solitary Electrons" describes a technique to infer the Wigner function of electrons emitted well above a Fermi sea. This function encodes all the single-particle properties of the electronic state and is of crucial importance when trying to distinguish between quasi-classical and quantum currents, the last ones being at the root of electronic quantum optics. Other schemes of quantum tomography are nowadays proposed and tested. The technique of this paper presents the advantage of being broadband and not limited in the phase space to energy and time close to the Fermi sea. It requires mastering high frequency excitation and detection scheme, which is well demonstrated. Despite being far away from the quantum regime here, the obtained resolution should be sufficient to enable quantum-limited measurements. I therefore think the impact of the paper to the community will be important. Yet, I have the next few concerns and invite the authors to revise their manuscript before making a final decision for its suitability to Nature Communications.

- 1) Is the sample used the same as in PRL 111, 216807 (2013) ? If yes please state it.
- 2) The technique is demonstrated for high-energy solitary electrons emitted from the electron pump. The author claimed that it is broadband. I agree for large emission energy. What would be the lower bound? Would this detection scheme works for other type of single excitation source (quantum capacitor for example) ?
- 3) The core of the paper is the description of the Wigner function reconstruction from measurements. It was difficult to understand how it is realized in the main text without reading the Methods part. I suggest that part of the methods is included in the main text. Does the Radon transformation introduce any bias or uncertainty in the reconstruction (smoothing...)? What are the limitations of such a tomography?
- 4) The based temperature of the dilution fridge is 100 mK. Is that a cryogenic limit or due to heating during the measurement process?
- 5) If heating is the dominant process would this technique be reliable to test quantumness of sources usually operating at lower temperature?

Technical questions and suggestions:

- the figure legends are short and not self-consistent to describe what is plotted/ some piece of information is missing on the figure :
 - o fig 1a) can't be understood without reading the main text and the methods.
 - o Fig 2a) what is the color code? What is represented on the left axis?
 - o Fig.3 b) should state that VG1 is on the left axis.
- The description of the sample is incomplete. I had to read a previous reference of some authors (PRL 111, 216807 (2013)) to get what VG2 describes.
- The main result of the paper is the tomography of excitations produced under different operating conditions. Why is the model in the supplementary information and not in the main text?
- In the conclusion, it's written "Our method can probe reveal...", one verb should be removed.

Reviewer #3 (Remarks to the Author):

The paper by J. D. Fletcher et al., entitled "Quantum Tomography of Solitary Electrons" reports on the tomographic measurement of a hot electron, whose energy is far above (~ 100 meV) the Fermi energy of the system, emitted by an on-demand single-electron source. The authors employ a GaAs based quantum dot and periodically drive it to realise a single-electron source. A quantum Hall like edge channel is used as an electron waveguide, where the confinement potential is set above the Fermi energy to deplete conduction-band electrons and to suppress energy relaxation due to phonon emission. In this system a potential barrier formed by a quantum point contact works as an energy-selective detector, where only an electron with energy above a certain threshold can pass through. The authors control the threshold with a linearly ramped AC gate voltage on top of a variable DC voltage. The authors collect the data while precisely controlling a delay of the AC ramp from a single electron emission and a slope of the ramp and reconstruct Wigner quasi-probability function following the method proposed by some of the authors [1]. The authors demonstrated that Wigner function of the emitted electron can be controlled by changing the emission condition. From the obtained Wigner function the authors calculated that the average and the maximal phase space density are 4% and 7% of the quantum limit which is partly explained by finite resolution effects. The authors also discussed the areal resolution in energy-time phase space of the method and claimed that it is sufficient to resolve a quantum-limited Wigner function.

Single-electron quantum optics is a relatively new field and attracts a particular interest in terms of quantum information as well as fundamental physical point of view [2-4]. There it is very important to analyse the quantum mechanical property of a single-electron source [5, 6]. The

method demonstrated here gives a useful new tool to analyse a single-electron source, where an emitted electron is isolated from Fermi sea. With the high quality of the experiment, where precise calibration and the idea to employ a controlled linear ramp of the detector voltage allow for the detection of an electron wave-packet with a time-domain width down to ~ 5 ps, I am convinced to publish the paper in Nature Communications with a minor revision. Before the publication I would like the authors to comment in the manuscript about the possibility to detect "quantum" feature with the demonstrated method. From the present data a single-electron source investigated here is "classical". Is there a possible revise of the source to observe "quantum" feature? Or is there another type of single-electron source, whose property can be analysed with the method demonstrated here and which shows "quantum" feature? I believe that comment on this will improve the impact of the paper.

I found one type in the paper.

1. Page 3, right column, around the middle: "the sweep rates near the point of \sim over a wide range (Fig. 3c squares)," should be " \sim (Fig. 3b squares),".

- [1] E. Locane et al. ArXiv: 1901.08940v1 (2019).
- [2] C. Grenier et al., Mod. Phys. Lett. B 25, 1053 (2011).
- [3] E. Bocquillon et al., Annalen der Physik 526, 1 (2014).
- [4] C. Bäuerle et al., Reports on Progress in Physics 81, 056503 (2017).
- [5] T. Jullien et al. Nature 514, 603 (2014).
- [6] A. Marguerite et al. ArXiv: 1710.11181 (2017).

Reply to the editor and referees for “*Quantum Tomography of Solitary Electrons*” by Fletcher *et al.* (NCOMMS-19-06893A)

Dear Editor,

We thank you and all three referees for their constructive comments on the manuscript. We are pleased that the main results, the reconstruction of the energy-time distributions of single hot electrons, are considered by the referees to be of high quality. We take it from some of their comments that we need to explain our expectations of ‘quantum behaviour’, and the results of this experiment carefully.

We set out our response below and have resubmitted a modified manuscript which clarifies what the limitations are of this experiment, for instance describing more explicitly the effects of classical ensemble averaging that is implied by the term ‘mixed state’. Hopefully this will remove any ambiguity about the results of this specific experiment versus the scope of the tomography protocol generally. We have also made modifications to address other technical comments as discussed below.

Kind regards

Jon Fletcher (on behalf of the co-authors)

Reply to referee #1

Referee #1 makes positive comments on the experimental results, but raises a question of what evidence for quantum behaviour we have in our results, saying:

“I think that the experiments as such are sound and of high quality. The text is also clear. However, I am exceedingly puzzled by the presentation that is, if I understood correctly, very misleading.

The name “Quantum Tomography” implies getting access to some quantum aspect of the electron. Quantumness is also implied by sentences such as “Our method can be used to test the degree of ‘quantumness’ of an electron source from measurements of the phase- space density” or “the resolution should be sufficient to enable quantum-limited measurements” although in the last sentence, the usage of the word “should” seems to imply that this goal has not been reached yet.”

We understand why referee #1 suggests that showing negative regions of the Wigner function would provide clear evidence of being in the quantum regime. Unfortunately, we cannot reveal such a ‘slam-dunk’ demonstration of ‘quantumness’ because substantial negative regions in the Wigner function should not exist for this source, even in the pure state case (regarding off-diagonal elements in the density matrix, see below). Our conditions [Ryu et al. PRL 2016, Kashcheyevs and Samuelsson PRB 2017] are rather far from the kind of extreme ‘two slit’/‘Schrodinger cat’ scenarios that result in negative values of the Wigner function, as seen in experiments with atoms [Kurtsiefer Nature 1997] or photons [Ourjoumtsev et al. PRL 2006]. There are many situations where the Wigner function is positive only, and even a perfect ‘quantum tomography’ would reflect this (textbook Gaussian wavepackets, ‘non-classical’ squeezed states of light [e.g. Breitenbach et al.

Nature 1997]). This means that we appeal to other arguments to consider where we are on the 'quantum to classical' (or 'pure to mixed' state) axis, using our measurements of the state purity.

Something to emphasize is that our approach is *compatible with both classical and quantum limits*, from a pure state to a mixed state representation. In contrast, we suspect that a classical argument alone would be subject to doubts regarding compatibility with a microscopic physical picture. The purely classical model, before blurring by resolution effects, is actually an infinitely narrow line in the in E,t plane, something which is questionable for time scales not so far from our experiment (which is in the picosecond not femtosecond regime, see note below). This is why we use a model that is compatible with a quantum picture from the outset, pushing further into the quantum regime with future experiments.

In our new submission, we state in the abstract and in the main text that there is a limitation imposed by classical fluctuations. We point out that we are in the limit of a mixed state 'ensemble' measurement at this point, but that we do not think that this is a fundamental limit. Modifications like increased ejection rates and reduced technical jitter will increase the measured state purity and we ultimately expect to resolve the quantum broadening directly. We might then envisage attempting experiments which reveal interference (e.g. ejection from two physical points, or from different orbital states), something which we added to the conclusion of the paper.

"On the other hand, unless I am mistaken - I would be happy to be contradicted - I feel that the observed data can be entirely understood at the classical level (e.g. the classical model of Fig. S4b looks quite good when compared to Fig.3) which is consistent with my understanding of what is going on."

A strictly classical treatment would require each electron to have well-defined energy and arrival time, and that the barrier would have a well-defined transmission threshold. Broadening would only be caused by fluctuations of these parameters. Our resolution in energy is ~ 0.8 meV and in time ~ 0.3 ps, which is of comparable size to the minimum uncertainty limit. Depending on the relative size of the uncertainty in energy and time of emission, quantum effects could become resolvable in one component if that in the other is small. Because we are so close to this quantum limit we feel it is not appropriate to apply a purely classical treatment. We do, however, need to clearly acknowledge that our results are affected by classical fluctuations in the text, as stated above.

"...measuring "quantumness" would require accessing the very fast oscillations of the wavefunction which occur at very fast time scale (10^{-14} seconds approximately for 100meV) which sounds very difficult with a setup that measures at frequencies smaller than 100 GHz."

This estimated time scale for oscillations in the wavefunction is too small as this uses the total energy (~ 100 meV) rather than the kinetic energy of drift motion [Kataoka et al. PRL 116, 126803 (2016)] (~ 2 meV). The actual time scale is closer to ~ 2 ps, nearer the achievable experimental controls. We also note that we do not need to provide an oscillating reference signal at this speed to resolve this as in optical homodyne schemes [Breitenbach et al. Nature 1997] or previous electron tomography experiments [Jullien et al Nature 2014]. In those cases a local oscillator (or modulated Fermi sea) provided a reference state related to the energy the particle being measured. In our case we are using the selectivity of transmission of different eigenmodes. The coherent superposition of these after the energy filtering barrier enables the conservation of wavefunction phase information on very fast time scales.

"Unless the authors can show that coherent off-diagonal elements of the density matrix are present (negative values of the Wigner function), there is no convincing evidence for "quantumness"."

We note here that the magnitude of off-diagonal elements of the density matrix is 'quantum-mechanical-representation-dependent' and is, generally, not equivalent to negativity of the Wigner function. While we cannot demonstrate negative regions in the Wigner function we do have off-diagonal elements. For example, coherence between wave-packet components at different energies (that is quantified by off-diagonal elements of the density matrix in the energy representation) is a necessary condition for the "chirp effect" we report, even if the Wigner function is positive-definite (but correlated) Gaussian. These off-diagonal elements are taken into account in our calculation of purity via diagonalization of the density matrix (see Supplementary section F). The inevitable presence of off-diagonal matrix elements even in a density matrix of a strongly mixed, classically-modellable state is one of the reasons why we report representation-invariant measures - the maximal single particle weight P_1 and the purity γ .

The referee suggests that we modify the manuscript regarding the 'quantumness' of the results:

"Hence, at minima, I recommend that the manuscript be striped from all the misleading quantum aspects (or that the authors provide convincing proof that a quantum treatment is needed to explain the data). It does not follow that the results are uninteresting, but a proper discussion of the new information conveyed on the classical probability distribution would be needed to assess this aspect."

We have revised parts of the text to clarify what should be read from our results, for instance, we now state in the abstract:

"We cannot yet resolve the pure state Wigner function of our excitations, because both source purity and detector fidelity are suppressed by classical fluctuations."

We hope that with clarifications like this (along with our above observations about the expected Wigner function of our source, the nature of the tomography technique itself and of the illustration of the relevant timescales) is sufficient to explain why presenting the data in the framework of a quantum tomography is appropriate.

Reply to comments of referee #2

Referee #2 comments positively on some key aspects of our paper, for instance that

"The technique of this paper presents the advantage of being broadband and not limited in the phase space to energy and time close to the Fermi sea. It requires mastering high frequency excitation and detection scheme, which is well demonstrated. Despite being far away from the quantum regime here, the obtained resolution should be sufficient to enable quantum-limited measurements. I therefore think the impact of the paper to the community will be important."

but asks us to address some technical issues as listed below.

"1) Is the sample used the same as in PRL 111, 216807 (2013) ? If yes please state it."

No, this is a different sample. We have fabricated and tested dozens of devices since then, testing many features of hot electron transport in great detail. Here the device is of same type as the structures used to test velocity of electrons [Kataoka PRL 2016] and is exactly the same device used to test phonon emission [Johnson PRL 2018]. We have added this note to the Methods text. Here, we are using only part of the device; the upper branch, which features a much longer path length between the source and the detector is not used. For the settings used here, none of the electrons

travel that path (these would be clearly seen as a peak in the arrival time distribution at a later time, as in [Kataoka PRL 2016]).

“2) The technique is demonstrated for high-energy solitary electrons emitted from the electron pump. The author claimed that it is broadband. I agree for large emission energy. What would be the lower bound? Would this detection scheme work for other type of single excitation source (quantum capacitor for example) ?”

The issue of applicability at different energies is partly addressed in section G of the supplementary information *“Generality of technique over energy and time ranges”*. This states some of the kind of practical limitations that one might encounter e.g. we cannot maintain an indefinitely long voltage ramp with our waveform, and the barrier must remain opaque to low energy electrons during the whole cycle. We speculate that the latter constraint, requiring that the AC barrier height modulation does not cross the Fermi energy, might pose a practical problem for very low energy sources [Bocquillon PRL 2012]. This can have the effect of enabling parasitic currents driven by either unbalanced thermal/technical bias voltages or unbalanced transients induced by the AC bias [Giblin et al. Journal of Applied Physics 2013]. On the other hand, because the overall energy scale is smaller for those devices, the size of the signals involved will be smaller (by a factor >100) and these effects may be less pronounced.

“3) The core of the paper is the description of the Wigner function reconstruction from measurements. It was difficult to understand how it is realized in the main text without reading the Methods part. I suggest that part of the methods is included in the main text.”

Unfortunately, reasons of space we had to put some technical details into the methods section. In the broader literature (beyond the subject of single electronic system) we have found that the technique of inverse radon transformation was quite common. The physical nature of our ‘projection plane is’ relatively unusual though, which is why we put this part in the main text and the details in the methods sections.

“[3 continued] Does the Radon transformation introduce any bias or uncertainty in the reconstruction (smoothing...)? What are the limitations of such a tomography?”

This is an interesting question. We can highlight three effects which are described in the supplementary information.

- i) Artefacts (e.g. ‘streaks’) generically appear in ALL numerical back projections which somehow reflect the angular ‘granularity’ of the projection intervals. For the high density of data we have here, this is a weak effect. We have added a note to highlight this.
- ii) The filtered backprojection process necessarily includes a high pass filter (this is an essentially part of the algorithm) which naturally amplifies experimental noise. This is attenuated very slightly by a band pass filter, which smooths very fine features, while leaving the gross ones unchanged.
- iii) Bandwidth limitations make a pure time ‘projection’ ($\theta = 90^\circ$) physically impossible to achieve. This prevents the detection of structure on time scales shorter than the barrier rise time, for instance very high chirp angles would saturate (this is not what is seen in fig 3d as we stay away from this limit).

“4) The base temperature of the dilution fridge is 100 mK. Is that a cryogenic limit or due to heating during the measurement process?”

The RF drive raises the mixing chamber temperature of the fridge from $T < 30$ mK to around 100 mK by the rf drive excitation**, which is what we quote (We have added a note to the text to indicate that this is the measurement state). We do not have a direct measurement of the electron temperature but expect the electron temperature to be somewhat higher than this. Generally, operation at slightly elevated temperatures does not have a strong effect on the loading of the pump (pumps work at $T > 1$ K) and we have performed many hot electron transport experiments in He3 cryogenic systems with base $T > 300$ mK. It is likely that in systems that operate close to the Fermi energy more precautions against heating would have to be taken, but a key point is that the overall energy scale is much lower, so we expect the drive signal losses to be much reduced.

**In fact, we have recently reduced RF losses in the system and can operate pumps with a mixing chamber temperature of only ~ 40 mK.

"5) If heating is the dominant process would this technique be reliable to test quantumness of sources usually operating at lower temperature?"

For our high energy electrons, we do not expect heating to be an issue in tests of quantumness. For lower energy sources, our level of heating might be a concern. However the energy scales are all smaller in that case, including the RF signal size required to eject each electron, so we expect a reduced power dissipation.

*"the figure legends are short and not self-consistent to describe what is plotted/ some piece of information is missing on the figure :
o fig 1a) can't be understood without reading the main text and the methods."*

We have expanded the figure 1a) caption.

"o Fig 2a) what is the color code? What is represented on the left axis?"

The colour scale is dI_T/dS , where dS is a normalised Energy, time coordinate (the integral of this always conserves the total amount of transmitted charge). We have changed the figure caption to reflect this.

"o Fig.3 b) should state that VG1 is on the left axis."

We have added the label.

" The description of the sample is incomplete. I had to read a previous reference of some authors (PRL 111, 216807 (2013)) to get what VG2 describes."

We have added a sentence to explain the role of V_{G2} .

"The main result of the paper is the tomography of excitations produced under different operating conditions. Why is the model in the supplementary information and not in the main text?"

If we had space, in principle we could move section D of the supplementary information to the main text – we have asked the Editor about this. We note that the effect of the upward trajectory of the electron emission energy has also been seen in other calculations, not just this semiclassical one [see Figure 2. of Kashcheyevs and Samuelsson PRB 2017], so the experimental result is really the new result, not this model calculation.

"In the conclusion, it's written "Our method can probe reveal...", one verb should be removed."

We have fixed this typographical error

Reply to comments of referee #3

Referee #3 comments positively on our manuscript:

“The method demonstrated here gives a useful new tool to analyse a single-electron source, where an emitted electron is isolated from Fermi sea. With the high quality of the experiment, where precise calibration and the idea to employ a controlled linear ramp of the detector voltage allow for the detection of an electron wave-packet with a time-domain width down to ~ 5 ps, I am convinced to publish the paper in Nature Communications with a minor revision.”

They ask for a comment about the prospect of observing quantum features.

“Before the publication I would like the authors to comment in the manuscript about the possibility to detect “quantum” feature with the demonstrated method. From the present data a single-electron source investigated here is “classical”. Is there a possible revise of the source to observe “quantum” feature? Or is there another type of single-electron source, whose property can be analysed with the method demonstrated here and which shows “quantum” feature? I believe that comment on this will improve the impact of the paper.”

It is not just the source that may need to be modified, as some of the broadening is attributable to the energy resolution of the barrier. We think that increased ejection rates will lead to an increased measured state purity (which we can directly quantify) as this diminishes the significance of the fixed energy resolution enabling use to reach closer to the quantum limit. We can also speculate about experiments scenarios which will reveal interference (e.g. ejection from two physical points, or from different states in the pump) but these will require substantial modification of the source, which is beyond the scope of this report on the tomography technique itself. An alternative method discussed in Locane et al. [<https://arxiv.org/abs/1901.08940>] would involve applying deliberately non-linear potential pulses on top of the linear ones. We have added a note to the text to reflect on this point.

Detailed list of changes

1. Revised abstract (fragment).

...We reconstruct the Wigner representation of the mixed-state density matrix for $\textit{solitary}$ electrons emitted by an on-demand source. This reveals highly localised distributions isolated from the Fermi sea. We cannot yet resolve the pure state Wigner function of our excitations, because both source purity and detector fidelity are suppressed by classical fluctuations. We can, however, test the degree of ‘quantumness’ of the source from measurements of the phase space density. We can also resolve some of the structure of the Wigner function, a ‘chirp’ controlled by emission conditions. This tomography scheme, when implemented with sufficient experimental resolution, will enable quantum-limited measurements...

2. Figure 1 new caption:

*$\textbf{Electron tomography scheme using a modulated barrier.}$
 \textbf{a} , An unknown Wigner distribution $W(E,t)$ of a periodic electron source electron can be filtered using a linear-in-time threshold energy barrier set at height E_T . The transmitted and reflected part, labelled P_T and $1-P_T$ result in a proportionate transmitted and reflected currents. A marginal projection of this distribution in the energy,*

time plane can be measured by fixing the ramp rate of the barrier β_E which sets E_T , then moving the threshold boundary along the axis ϵ in increments $d\epsilon$, while measuring the resulting changes in transmitted current. Repeating the experiment at different ramp rates (which sets the angle θ) gives enough information for a numerical reconstruction of the distribution.

\textbf{b}, False-colour scanning electron micrograph of device identical to that measured (see methods for details). The electron pump (left, highlighted green) injects pump current I_P . The barrier (right, highlighted red) selectively blocks electrons giving transmitted current $I_T \leq I_P$. The path between these is indicated with a line. The gates along the path (controlled by V_{G4}) depletes the underlying electron gas but do not block the high energy electrons (see supplementary materials).

\textbf{c}, Typical time-dependent control voltages for pump V_{G1} and probe barrier V_{G3} (each has a DC offset - see methods).

\textbf{d}, Electron potential $U(x)$ along the electron path between source and probe barrier at three representative stages for pumping (left) and blocking (right)

3. Definition of VG2 added to text:

The right hand barrier, controlled by V_{G2} determines the number of electrons pumped, and linearly controls the ejection energy [Fletcher et al. PRL 2013]

4. Note added after introduction of first Wigner function reconstruction:

As some classical fluctuations are present in our experimental implementation we interpret our results as an effective mixed state Wigner function [Locane et al. Ref. 26]. Our results are therefore an ensemble measurement of many pure states, with a lower phase-space density than a pure state (discussed below). We can, however, resolve a feature likely to be common to each pure state, a feature of the distribution that we show can be controlled by electron ejection conditions.

5. Revised Figure 2 caption:

\textbf{a} Angle dependent projection of single electron density.

\textbf{a} Colour plot: projected electronic density (sinogram) at various angles θ . Colour scale corresponds to $dI_T/d\epsilon$ where $d\epsilon$ is a normalised energy, time step (the integral always give the pump current I_P). Depending on θ , the projection axis ϵ can correspond to energy, time or a mixture. Upper panels: selected marginal projections at angles where ϵ corresponds to a time projection (i) energy projection (ii) and an mixed projection (iii).

\textbf{b} Inverse Radon transform of the data in *\textbf{a}* giving the Wigner phase-space density in units of \hbar^{-1} .

6. Figure 3: a “ V_{G1}^{DC} ” label has been added to panel **b**

7. Note added to methods section B:

This is the same kind of device as used to measure electron velocity

\cite{kataoka2016timeflight} and phonon emission \cite{JohnsonPRL18}.

8. Note added to supplementary section B:

Numerical backprojection schemes can display artefacts (e.g. streaks) if the discrete gridding and angular resolution are not high enough (this is a weak effect here).

9. Word ‘probe’ deleted from summary

10. Figure label changed from ‘e’ to ‘b’ in sentence ‘tuned over a wide range (Fig, 3 b squares)’

11. Added text in methods section about base temperature: ‘with RF signals turned on’

12. Added text to conclusion:

It should also be possible to use this scheme to detect coherences via negative fringes in the

Wigner function arising from interference effects[26,36]; the characteristic oscillation period estimated from the kinetic energy of drift motion[25] is $\sim 2\tau_{ps}$, close to our accessible bandwidth.

REVIEWERS' COMMENTS:

Reviewer #2 (Remarks to the Author):

I thank the author for their answers to my comments and subsequent changes to the paper. My opinion is that the paper now reflects a high quality experimental demonstration of tomography.

Reviewer #3 (Remarks to the Author):

With the response and the revisions by the authors, I believe that the manuscript can now be published